# Automatic Monitoring of Relevant Behaviors for Crustacean Production in Aquaculture: A Review

**DOI:** 10.3390/ani11092709

**Published:** 2021-09-16

**Authors:** Daoliang Li, Chang Liu, Zhaoyang Song, Guangxu Wang

**Affiliations:** 1College of Information and Electrical Engineering, China Agricultural University, Beijing 100083, China; chang_liu@cau.edu.cn; 2National Innovation Center for Digital Fishery, China Agricultural University, Beijing 100083, China; 3Beijing Engineering and Technology Research Centre for Internet of Things in Agriculture, China Agricultural University, Beijing 100083, China; zhaoyang@cau.edu.cn; 4China-EU Center for Information and Communication Technologies in Agriculture, China Agricultural University, Beijing 100083, China; gxwangmail@163.com

**Keywords:** aquaculture, crustacean behavior, acoustic technology, machine vision, movement sensor

## Abstract

**Simple Summary:**

Automatic behavior monitoring, also called automated analytics or automated reporting, is the ability of an analytics platform to auto-detect relevant insights—anomalies, trends, patterns—and deliver them to users in real time, without users having to manually explore their data to find the answers they need. An analytics platform with automated behavior monitoring uses algorithms to auto-analyze datasets to search for notable changes in data. It then generates alerts at fixed intervals or triggers (thresholds), and delivers the findings to each user, ready-made. In-aquaculture scoring of behavioral indicators of aquatic animal welfare is challenging, but the increasing availability of low-cost technology now makes the automated monitoring of behavior feasible.

**Abstract:**

Crustacean farming is a fast-growing sector and has contributed to improving incomes. Many studies have focused on how to improve crustacean production. Information about crustacean behavior is important in this respect. Manual methods of detecting crustacean behavior are usually infectible, time-consuming, and imprecise. Therefore, automatic growth situation monitoring according to changes in behavior has gained more attention, including acoustic technology, machine vision, and sensors. This article reviews the development of these automatic behavior monitoring methods over the past three decades and summarizes their domains of application, as well as their advantages and disadvantages. Furthermore, the challenges of individual sensitivity and aquaculture environment for future research on the behavior of crustaceans are also highlighted. Studies show that feeding behavior, movement rhythms, and reproduction behavior are the three most important behaviors of crustaceans, and the applications of information technology such as advanced machine vision technology have great significance to accelerate the development of new means and techniques for more effective automatic monitoring. However, the accuracy and intelligence still need to be improved to meet intensive aquaculture requirements. Our purpose is to provide researchers and practitioners with a better understanding of the state of the art of automatic monitoring of crustacean behaviors, pursuant of supporting the implementation of smart crustacean farming applications.

## 1. Introduction

Aquaculture has become one of the largest commercial and economically important industries in recent years [1]. Lobsters, crayfish, crabs, crayfish, prawns, and shrimp are the most valuable crustacean species groups with significant production. Shrimp and prawn catches recorded new highs in 2017 and 2018 at over 336,000 tons [2]. In aquaculture, most of the modern information technologies are applied to production management and reliable monitoring of crustacean behavior is very important for aquaculture industries because it provides a starting point for welfare assessment [3,4]. Traditional crustacean behavior monitoring is mostly based on manual measurement. However, manual monitoring is usually laborious, time-consuming, and ineffective which thus limits its economic benefits [5,6].

Modern-day crustacean aquaculture originated in Japan in the 1930s [7], and automatic monitoring methods were developed in the 1970s and expanded rapidly around the world [8]. Automatic behavior monitoring in aquaculture is defined as the application of process engineering principles and techniques to precision fishery farming to automatically monitor and recognize animal behavior [9,10]. Until now, scholars and researchers have developed various automatic methods to monitor crustacean behaviors in laboratories or ponds, including acoustic technology [11], machine vision [12], and movement sensors [13,14]. Compared with the environmental parameter detection system, automatic behavior monitoring is a posteriori indicated, but it is very meaningful for welfare evaluation [15,16]. In terms of feeding, moving, home range, and activity, rhythms may grasp biological behavior information, monitor animal health in real time, and provide early warning of diseases [17]. Therefore, real-time monitoring of individual behavior is important for improving production in crustaceans, and there is an urgent need for farmers to monitor behavior in real time, which allows fishermen to take actions in the initial stages of welfare or disease problems to meet the intensive aquaculture requirements [18].

This paper aims to summarize the characteristics of different crustacean behaviors and various automatic aquaculture behavior monitoring methods that have been used over the past three decades. In addition, this article also discusses and summarizes the advantages and disadvantages of each method. Finally, we present potential applications and new techniques for the automatic monitoring of crustacean behavior and the major obstacles that need to be overcome. This review could provide a valuable reference to guide future research into intelligent technologies for behavior monitoring and help practitioners to assess crustacean welfare.

## 2. Important Behaviors in Crustacean Aquaculture

Modern technology offers the possibility for real-time shrimp behavior monitoring in aquaculture as a fast and automatic research topic and a repeatable method [19]. In general, when shrimps are in different physiological states, their behavioral profile will change, such as posture, sound frequency, and activity rhythms [20,21,22]. Figure 1 shows the number of papers related to different methods and monitoring behaviors. The most popular methods are acoustic technology, machine vision, and movement sensors. Notably, feeding behavior, movement rhythms and reproductive behavior are the main focus of automated monitoring methods. We will focus on understanding the characteristics and influencing factors of the above three behaviors, which can provide a basis for the development of various automatic monitoring methods.

### 2.1. Feeding Behavior

Feeding is the primary factor for determining the efficiency and cost of aqua feed, which may represent a considerable proportion of the crustacean farming budget [23]. Crustaceans use visual, mechanoreceptor, and chemoreceptor systems to detect the location of food sources, and when food is available, crustaceans change their sound signatures and movements [24]. Feeding behavior can reflect many aspects of an individual organism. The survival rate and molting cycle of red swamp crayfish are associated with different feeding rates [25]. Santos et al. revealed that white shrimp display nocturnal feeding and locomotor rhythms [26]. Thus far, scholars have only used computer vision and passive acoustics to recognize feeding behavior. In the future, we can focus on variables that reflect feeding behavior, such as activity rhythms, posture, and position, and use more types of sensors to indirectly monitor feeding behavior. Feeding table and schedule is a common and accurate feeding method, but automated feeding behavior recognition also has great potential in determining when to start and stop feeding in order to improve feed conversion rate and reduce costs [27].

### 2.2. Movement Rhythms

In addition to feeding behavior, movement also plays a major role in determining the structure of populations and communities, as well as the evolution and diversity of life [28,29]. Movement rhythms are defined as the recurrence of any event within a biological system at more or less regular intervals. Crustacean movements can be categorized spatially as homing, nomadic, or migratory, and temporally as daily, ontogenetic, or seasonal [30,31]. As a basis for welfare assessment, the rhythms of movement behavior can only be used to help choose the correct location for fishing, and are not the core of assessing performance under aquaculture conditions. Therefore, monitoring of movement rhythms during the fishing season will be of great help for choosing the correct location for fishing. Studies designed to understand crustacean behaviors have used techniques such as tag–recapture [32,33], visual tracking [34,35], acoustic telemetry [36,37], or a combination of some of these techniques [38]. However, differently from land organisms, marine species present technical difficulties when their movement is monitored over prolonged period of time, due to the presence of saline water [39,40].

### 2.3. Reproductive Behavior

The reproductive behavior of animals is a significant manifestation of their life and mating is a key step in reproduction [41,42]. The mating process includes approach, touch, mount, turn, rolling, and thrust [43], and this process lasts between 28 s and 6.40 min [44,45]. Obvious external action characteristics and behavior duration are the basis for monitoring using automated methods. Monitoring reproductive behavior can accurately determine when mating occurs, and guide fishermen to perform artificial insemination, thereby increasing reproductive yield. In addition, by monitoring whether the reproductive behavior is normal, disease monitoring and prevention can also be effectively carried out. Therefore, analysis of reproductive behavior can effectively improve crustacean production and larval quality. However, many factors influence reproduction, which are broadly divided into temperature effect, photoperiod effect, and season effect [46]. In addition to environmental effects, individual-level factors also affect reproduction, such as body size, the history of sex, investment in offspring, fitness, and dominance status [47].

We can only effectively analyze behaviors that have been monitored by automated methods. There are also some behaviors such as struggle behaviors that reflect changes within the population, but understanding these behaviors is limited by the inefficacies of manual observation. Under intensive cultivation conditions, the accuracy and precision of the monitoring results are dependent on multiple factors spanning individuals, the environment, water quality, and device model [48]. Based on the studies discussed above, we can appreciate that crustacean behaviors are complex and difficult to monitor. Discussed below are the current technical shortcomings and the future development direction, which suggest a pressing need for providing new ideas for further improvement of intelligent monitoring for farmers and information technicians.

## 3. Behavior Monitoring Methods Based on Acoustic Technology

Autonomous acoustic monitoring is a technique using sound waves to remotely measure information. Acoustic technology has been widely used in species identification [49], biomass estimation [50], and behavior monitoring without causing stress to crustaceans [51]. For underwater monitoring, acoustic technology has key advantages over light waves and electromagnetic waves because of the long propagation distances [16]; another advantage of acoustic technology is that its measurement results are less affected by water turbidity and underwater light [52]. According to data acquisition methods, acoustic technology can be divided into passive acoustics and active acoustics. Active acoustics includes sonar, echo, and acoustic telemetry. Sonar and echo technology are more used to measure the density of crustaceans, and acoustic telemetry is more common to monitor crustacean behaviors.

### 3.1. Passive Acoustics

According to Howe et al. [11], passive acoustics is the action of listening for sounds, often at specific frequencies or for purposes of specific analyses. The basic technique involves using one or more hydrophones or appropriate acoustic processing systems to detect natural vocalizations made by underwater creatures. However, the frequency of the sound is very broad. Therefore, the hydrophones in the passive acoustic system placed into farm ponds will be equipped with an amplifier attached to a digital acquisition unit [53]. The digital acquisition unit is attached to a personal computer, which is used to provide valuable information and this process is often undertaken by complex and specific algorithms [54]. Many investigations have indicated that when some behaviors occur, crustaceans emit different sound frequencies, including feeding [55], mating [56], carapace vibrations [57], snap [58], and stick and slip friction [59,60,61]. With such variety of sound production mechanisms, the characteristics of the sounds produced by crustaceans are diverse [62,63]. According to the above theoretical basis, experts can identify crustacean behaviors via long-term acoustic monitoring of sounds.

The mechanisms and spectral characteristics of crustacean behaviors are heterogenous. In terms of feeding sounds, the physical production mechanism is that shrimp use mandibles and maxillae to tear feed pellets into pieces before entering the oral cavity [63]. Some scholars have used the sound spectral features of feeding as an indication of pellet consumption [64]. These experimental results show that the correlation between sound and feeding behavior can reach more than 95%. Although passive acoustic technology can provide guidance for measuring the relative intensity of feeding activity, it is unclear how accurate it is at estimating the quantity of consumed pellets from feeding sounds. In terms of activity rhythms, calibrated hydrophones can be used to measure the relationship between crustacean sound signals and intraspecific interactions (encounter/approach, fighting, and successive tail flips), circadian rhythm [65], and seasonal rhythm [66]. If the *p*-value of the significant difference between activity rhythm and sound is less than 0.05, it fully proves the reliability of using passive acoustic technology to monitor crustacean activity rhythm. In addition to monitoring activity rhythms, Kikuchi et al. also found that the frequency of stridulating sounds from Japanese spiny lobsters tended to increase at night with the degree of tidal change, and that they are more active during large tidal changes [67]. Bohnenstiehl et al. showed that sound pressure levels were positively correlated with snap rate (r = 0.71−0.92) and varied seasonally by 15 decibels in the 1.5–20 kHz range [21]. The activity rhythm and snap information measured by passive acoustic technology can provide guidance to determine crustacean distribution and optimal harvest time. Commonly cited advantages of passive acoustics include that they can rapidly and noninvasively sample large crustacean volumes. However, other forms of impulse may be similar to the characteristics that could potentially be misclassified as specified behavior, which is also the main cause of error [53]. Therefore, the key challenges are improvements in automated signal detection and classification. The signal detection method based on machine learning can extract the time–frequency characteristics of the sound and filter out the interference of noisy sounds. In addition, post-processing and analysis of large datasets are current difficulties [68,69]. In order for passive acoustics to be better applied to crustacean behavioral monitoring, specialists can use big data technology to achieve intelligent data processing and analysis and develop user-friendly software that can be used by fishermen and ecologists.

### 3.2. Acoustic Telemetry

Acoustic telemetry is technology to transfer information underwater using sound; it was first used in the early 1970s and has been continuously improved over time [70,71]. Figure 2 is a schematic diagram of acoustic telemetry. An acoustic telemetry system designed specifically for aquaculture includes an acoustic receiver with hydrophones, radio smart transmitters, tags, and base station with antenna and computer [29]. Hydrophones are usually mounted on surface buoys, which listen to the tagged animals [72], and an acoustic transmitter sends out information, e.g., an ID code, as short tone-bursts, which are picked up, decoded, and timestamped by an acoustic receiver [73]. Finally, the radio sends tag information and a time stamp to the base station. Commonly, the base station analyzes the arrival time of different signals to determine the location of the underwater animals; this information consists of presence, movement, and behaviors of the tagged animal [74]. Therefore, this method is effective for estimating daily home ranges, core areas of activity [75], nomadic movements [76,77], activity patterns [78], and distance traveled, as well as behaviors [79] such as feeding, molting, and reproduction. It is worth noting that this technology cannot accurately gauge local movements.

As the most critical step of monitoring crustacean behavior, individual data concerning underwater animals collected by acoustic telemetry are very important for fishermen and researchers, and the range is from small ponds to large lakes and coastal areas [73]. Compared with radio and PIT-tag telemetry, acoustic telemetry is more effective for tracking aquatic organisms in both estuaries and oceans [71]. For different crustacean behaviors, the acoustic telemetry monitoring system also has subtle differences in data processing and analysis. Due to the maturity and completeness of the equipment, many scholars have used Canadian VEMCO-brand (Halifax, NS, Canada) acoustic telemetry systems with tags, which are one of the most widely used systems to obtain data on crustacean positions. The position information can be directly quantified into diurnal activity rhythms [29,80,81,82], seasonal movements [83], home range [31,75,84], nomadic behavior [72,76], and migratory patterns [85,86,87]. Compared with passive acoustic monitoring, acoustic telemetry technology is more effective in determining the activity pattern of individual crustaceans. In addition to activity rhythms, VEMCO VR2 systems have been used to reveal that female lobsters’ reproductive migration occurred between 5 June and 25 August. This result provides reference for revealing the reproductive behavior of the lobster [31,79]. Information such as this can guide fishermen to carry out artificial breeding in time or create suitable natural mating environments to improve breeding production.

In summary, all the above studies show that acoustic telemetry can monitor aquatic animals in a free-living state with the advantage of location. The detailed information concerning crustacean behaviors derived from acoustic technology studies is listed in Table 1. Of course, acoustic telemetry also faces some difficulties and challenges. A common concern is the potentially adverse effects on animal survival and behaviors. The difficulty is that in order to obtain behavior data, the animal to be monitored must be tagged. In addition, telemetry projects are often relatively expensive. Although the acoustic receivers and base station can be used repeatedly, tags are usually considered expendables [88]. Compared with passive acoustics, fewer crustaceans tend to be monitored and tracked in acoustic telemetry contexts. The data resolution has tended be very high, but some complex behaviors such as sublime aggression, courtship, and some actions that are transmitted by chemical signals are hard to identify by acoustic telemetry. Another technical difficulty is quantifying spatial location. This represents an important focus area for future research and development. By combining the Internet of Things, artificial intelligence, and cloud computing, it is possible to identify the spatial position information of crustacean movements pursuant of intelligent optimization and decision-making control functions in smart aquaculture.

## 4. Behavior Monitoring Based on Machine Vision

Underwater machine vision technology has been used since the 1950s to study the behavior, distribution, and abundance of marine and freshwater organisms [89]. Applications of machine vision have increased considerably in two major aquaculture domains, namely: (1) pre-harvesting and growth of underwater animals and (2) post-harvesting [90]. This technology can provide an effective means for the analysis of individual features [91,92], species classification [93], vocalizations [94], and behavior recognition within complex data sets at scales and resolutions not previously possible [95,96]. Machine vision technology can help us solve some important problems concerning ecology, social structure, collective behavior, communication, and welfare [97]. It can also save initial raw information for potential re-analysis, and record both visible benthic organisms and other biological activity [98]. Machine vision methods can quantitatively analyze behavior and greatly increase the efficiency, repeatability, and accuracy of image review, which is also a prominent advantage compared to acoustic technology. The typical equipment includes an industrial camera, source, acquisition card, and image processor. Based on the different wavelengths utilized by cameras, light can be divided into visible and infrared. The system structure and monitoring flow chart which utilizes visible light as the light source is shown in Figure 3.

### 4.1. Machine Vision Based on Visible Light

Machine vision technology based on visible light is widely used for crustacean behavior monitoring compared to other types of light sources. Extant studies on the monitoring of shrimp behavior can be divided into two categories. Direct methods use the measured videos or images to obtain the feature, trajectory, angle, velocity, and range of crustacean activities, as well as other parameters. With indirect methods, crustacean behavior is monitored from information on uneaten pellets recorded by a camera.

#### 4.1.1. Direct Behavior Monitoring

Studies have shown that crustaceans exhibit particular behaviors in different physiological states [99]. According to the specifics of the experimental environment and the characteristics of action occurrence, image processing systems usually approach this by the applicable algorithms, including image preprocessing, image segmentation, and feature extraction. There are three major branches of image preprocessing, namely image reconstruction, image restoration, and image enhancement [100]. This involves many methods such as linear transformation, histogram equalization, filtering, increasing, and frequency domain enhancement [101]. Especially for aquatic creatures such as crustaceans, which can easily cause water turbidity, image preprocessing is commonly applied to improve the quality of turbid images. Due to the temporal and spatial characteristics of video images, the main idea of the moving target detection method is to extract the changed regions from the background in the video image [102,103]. In recent years, more and more methods have been proposed to provide accurate and consistent segmentation for moving target extraction; commonly used methods include threshold segmentation, region segmentation, and edge detection [104,105]. Analysis and extraction of target features is the final step of behavior identification of moving targets, involving color features, texture features, geometric features, and motion characteristics.

Crustacean behaviors are related to their size, shape, speed, and color. The appearance detection method can identify static visual appearance features that are lacking in motion-based technologies, and it performs well in a stationary scene where crustaceans exhibit minimal motion. Feature extraction from texture is a basic approach for identifying behavior, e.g., the texture feature was used to extract the patterns of bay lobsters’ exoskeletons to automatically classify the molting stage with a maximum accuracy of 98.61% [106]. Oishi et al. successfully detected shrimp mating motions using cubic higher-order local auto-correlation (CHLAC) features in conjunction with a subspace method, with a standard deviation of 5.8 ± 1.3 [107]. Although posture analysis based on skeleton characteristics is often used in agriculture for large animals such as cattle and pigs, Yan and Alfredsen also extracted the lobster skeleton to quantify the posture of the lobster; the migration of this technology will provide more technical support for the application of skeleton feature extraction methods in aquaculture [18]. Machine vision monitoring methods are also widely used in the measurement of crustacean movement rhythms. The working principle is that the visual monitoring system obtains the pixel coordinates of the crustacean according to the position of the crustacean in the image. The computer then converts the pixel unit to the actual distance (mm) according to the x, y Cartesian coordinates recorded by the tracking software. The researchers calculate the Euclidean distance between the coordinates to obtain the total distance traveled by each shrimp [12]. Using the methods mentioned above, Aguzzi et al. found that the measurement of displacement of lobsters displays diurnal activity rhythms and burrow-related behavior [93]. Crustaceans offer the benefits of delineated developmental life stages and the accumulation of environmental toxins which change their behavior [108]. Therefore, some scholars have been able detect changes by tracking and analyzing the locomotion behavior of shrimp exposed to toxic chemicals in the environment, especially their movement speed, which yielded a *p*-value less than 0.08 [109,110]. In addition to image processing algorithms, mature video behavior monitoring software platforms have been developed in recent years. Commercial software uses graphics and mathematical methods to describe motion trajectories [111,112]; a real-time monitoring system can analyze multiple behaviors at the same time [34,35], and an underwater imaging system has been independently developed by researchers [113]. Real-time monitoring systems can be operated in aquaculture over the long term to cover the entire crustacean life cycle, from nursery to fishing, identifying the growth status of crustaceans in real time, and providing early warnings and alarms for abnormal behaviors in a way which minimizes labor inputs. However, the high price is the main issue that restricts it from being widely used in aquaculture.

#### 4.1.2. Indirect Behavior Monitoring

In addition, other information can be used to indirectly quantify crustacean behavior. Uneaten pellets and displacement represent important information for analyzing, identifying, and monitoring shrimp behavior. Therefore, such methods can be used to quantify particular behaviors that are difficult to detect [114].

Detection of uneaten pellets is another way of using machine vision to monitor feeding behavior. During this process, the corresponding area and other parameters of the food pellets can be used to indirectly monitor feeding behavior [113]. Those authors also measured organic matter residues in pond sediments to estimate feeding behavior at night time. The remaining pellets can be used as an indicator of the feeding intensity, thereby saving the amount of feed and effectively reducing pollution in culture ponds, but the accuracy of the results cannot be quantified [115]. Although indirect information can be used to monitor behavior, compared with the direct monitoring method, it is less accurate and prone to errors. This information can also be stored in a big data database, and it can help information technology staff build an expert farming system. Long-term underwater imaging and expert systems can also help in terms of smart feeding decisions, smart sewage decisions, and abnormal status warnings. In summary, machine vision technology has improved task performance (automatic monitoring) in achieving a task (e.g., classifying images) from image data. This technology can be a highly reliable and accurate method for objectively measuring activity levels in aquaculture with a low consumption of labor and time. However, regardless of whether direct or indirect measurements are used, it is still in the experimental stage, and large-scale applications still need to overcome many practical problems. Crustacean activities are mainly concentrated at night; the water can lead to reflections and the dark surface of the shrimp will directly reduce the clarity of the acquired video images. In addition to these practical problems, machine vision technology also faces some challenges in monitoring shrimp movements. The complexity of the monitoring environment and the uncertainty of the monitored objects are the biggest factors that interfere with shrimp behavior monitoring. There is an urgent need to improve the technology for extracting moving targets in underwater video images, and software to analyze more specific behaviors will become more important in the future.

### 4.2. Machine Vision Based on Invisible Light

Invisible light is an electromagnetic wave that cannot be seen by humans. The wavelength range is greater than 760 nm and less than 380 nm. The principle applied in aquaculture is based on the absorption of invisible light in water, resulting in variable brightness, which is not affected by visible light intensity and can yield good imaging results in dark places such as inside animal shelters [116,117]. Most crustacean species are nocturnal, remaining inside shelters during the day and actively foraging outside at night [118]. Therefore, invisible light technology is more suitable for capturing dim images of shrimp at night than visible light technology. Due to the low cost and low requirements of visible light intensity, it has a unique ability to fully understand the behaviors and rhythms of shrimp in aquaculture, which has poor lighting.

Invisible light technology provides a new method for accurately identifying crustacean behavior and mainly includes infrared imaging technology and X-ray imaging. The advantages of using infrared imaging technology to monitor crustacean behavior, including the fact that crustacean eyes are not sensitive to the infrared light used in the system and the scattering of infrared light in water does not tend to present a problem [117]. However, the major disadvantage of infrared light is that the attenuation coefficient and absorption of light in water increases dramatically as the light wavelength increases into the visible red region and then increases exponentially in the infrared region [119]. Hesse et al. used infrared photoelectric sensors to collect infrared images and study the different reactions of lobsters when different predators approach [120]. Ahvenharju and Ruohonen used ballotini glass beads to label diets with X-ray, and the number of ingested glass beads in the digestive track was counted from the X-ray images [121]. The accuracy rate was 92.8 ± 8.6% and the results confirm that using an X-radio graph technique makes it possible to measure the individual food consumption of freshwater crayfish juveniles reared communally. Invisible light technology is not affected by the light of the aquaculture environment, which can monitor crustaceans at night.

Invisible light has been used for monitoring crustacean behavior in laboratories and ponds. Compared with visible light systems, invisible light imaging technology requires no calibration, and is more suitable for measurements in turbid water with complex light conditions. In addition to behavior monitoring, it has also been used in aquaculture biomass estimation, 2D and 3D tracking, positioning of crustacean stocks, and various behavioral analyses [122]. However, further research is needed before such technology can be applied to commercial aquaculture to obtain real-time data on crustacean behavior pursuant of minimizing the interference caused by absorption, refraction, and scattering. Therefore, there is a need to improve the ability of invisible light technology to monitor crustaceans under high illumination levels or longer distances. For both visible light and invisible light methods to monitor the movement of crustaceans, improving image feature extraction technology and solving the problem that machine vision technology cannot be applied in high-density and large-population breeding contexts are still challenges to be overcome.

Overall, monitoring crustacean behavior through images with machine vision technology is currently an important application and research focus for realizing precision aquaculture, and the detailed information concerning machine vision technology for crustacean behavior monitoring is listed in Table 2. However, most research is still at the laboratory stage, and this method is not suitable for detecting some inconspicuous behaviors and behavioral transmission between crustaceans based on chemical signals which also cannot be detected. Therefore, it is necessary to develop a high-resolution monitoring system capable of local amplification.

In summary, future research and development directions can be demarcated as follows: (1) The focus on spatiotemporal and spatial sequence will continue to improve the accuracy and robustness of machine vision recognition of crustacean behavior. It is expected that algorithms similar to two-stream networks and 3D convolutional networks will be developed that can account for spatiotemporal sequences to achieve higher performance vis à vis shrimp behavior recognition methods. (2) The embedded vision system has the characteristics of compact structure, fast processing speed, and low cost; this is an important direction for the development of machine vision systems in the future. It also makes it possible to combine machine vision systems for large-scale popularization in aquaculture. (3) Machine vision systems that incorporate multiple technologies are also current and future research hotspots. For example, the fusion of the machine vision system and the Beidou navigation system can achieve high precision and low cost in the context of farmland navigation systems; the multiple video systems can collect more behavior information from crustaceans. The combined use of bio-floc technology and machine vision technology can make it possible to identify individual animals in intensive high-density environments.

## 5. Electrosensors

In addition to acoustic and optical technology, other sensors based on different parameters have been leveraged to identify and monitor crustacean behavior [125]. More broadly, sensors are often used for farming purposes. In recent years, more and more sensors suitable for crustacean behavior monitoring have been developed, and some equipment has been proposed for monitoring in aquaculture.

### 5.1. Accelerometer

Accelerometers are electromechanical devices designed to measure acceleration forces caused by gravity and the moving or vibrating activity of a subject. In particular, three-axial accelerometers can measure the motion, vibration, and displacement of underwater animals in X, Y, and Z directions [126]. When crustaceans undergo behavioral changes, they are usually accompanied by changes in movement speed or acceleration. Therefore, the high correlation between accelerometer data and movement of free-living individuals in different behavioral contexts is the key to identifying and monitoring different behavior states [127]. The development of accelerometer data loggers has made it possible to monitor daily patterns of behavior in many crustacean species, mainly lobsters, including slipper, spiny, and clawed.

Accelerometers are very effective in monitoring the activity rhythm of crustaceans, and are currently one of the main application areas of acceleration sensors. The collected accelerometer outputs can be converted into distances moved per unit time and scholars can estimate the distance moved by shrimp in a period of time according to this method to an extent that is statistically significant, that is, *p* < 0.005 [125,128]. However, the correlation between the movement and accelerometer is uncertain. Jury et al. obtained the value of r^2^ between the video activity and accelerometer activity of 0.898. The activity was defined as forward, backward, or sideways locomotion (>2 cm) for each lobster [128]. Goldstein et al. obtained the value of r^2^ between the video distance and acceleration of 0.53 and 0.63 [125]. These results indicate that the accelerometer can only estimate activity, and it is still not accurate enough for more demanding distance calculations. The system structure of monitoring crustacean behavior with sensors is shown in Figure 4; the acceleration sensor usually needs to be fixed on the crustacean, so it will apply pressure and thus cannot be used on small crustaceans (e.g., krill and fairy shrimp) This is also one of the challenges faced by intrusive automated monitoring methods. In addition to pure activity discipline assessment, some researchers have constructed models based on acceleration data and estimated the physiological status and welfare of crustaceans [13,14]. Thus, acceleration sensors can effectively be used to monitor the relative activity of lobsters over long periods in the laboratory and field, and this technology provides important reference information for the development of intelligent decision systems.

The accelerometer is similar to acoustic technology in that high turbidity and changing light levels will not affect the recording, and the sensor systems are readily adaptable to the field. The distance traveled and rate of movement can be estimated by calibrating accelerometer outputs compared to actual movements. Therefore, once appropriately calibrated, accelerometry appears to be a suitable method for assessment of movement patterns and distance traveled by animals above a certain size.

### 5.2. Electromyography

Electromyography (EMG) is an electrodiagnostic automated technique for evaluating and recording the electrical activity produced by skeletal muscles. An electromyograph detects the electric potential generated by muscle cells when these cells are electrically or neurologically activated. The structure of the system using EMG to monitor shrimp behavior is shown in Figure 5. The signal collected by EMG is converted and transmitted to computers, and the signals can be analyzed to detect physiological abnormalities, activation level, recruitment order, and the biomechanics of crustacean movement [129,130]. Therefore, these electrical signals can be used to design automated growth monitoring systems and develop intelligent decision-making and control systems for aquaculture.

According to some studies, the behavior of crustaceans can cause muscle cells to generate electrical potential [131,132]. EMG has been used to monitor the feeding behavior of individual crustacean. Gripping action is a key element of the feeding behavior of crustacean, especially lobsters. Therefore, the recorded electromyogram of the lobster claw muscle can characterize feeding behavior to an extent that is statistically significant at the 0.05 level [133,134]. It is reliable for monitoring feeding behavior and estimating the intensity of behavior based on chemical and biological EMG methods to gain a deeper understanding of crustacean feeding status, and the obtained data can be used to establish accurate growth models. The principle of studying lobster movement patterns is similar to monitoring eating behavior; the difference is that chronic electrodes are implanted on the shrimp’s legs instead of on the claws [129]. More importantly, the EMG pattern can be analyzed to determine whether the behavior is reflexive or spontaneous [130,135], which solves the problem that machine vision and acoustics cannot monitor some behaviors that are not obvious.

The studies above have shown that these sensors are highly accurate in detecting motion states and have great potential for estimating the intensity of behavior. Table 3 shows detailed information on sensor technologies. However, sensors need to be in contact with the crustacean, or even implanted in the crustacean during the measurement, which is an interventional monitoring method. The pressure and interference caused by this on the crustacean is difficult to gauge. In the future, miniaturized, lightweight sensors have great potential for reducing the pressure in small-scale biological monitoring contexts. Recently, namely, the fusion of acceleration sensors and other sensors (such as pressure, GPS, and acoustic tags); this has successfully been applied to monitor the ecology, physiology, and behavior of different fish [136]. This method of multi-information fusion could also be used in crustacean behavior monitoring. For behaviors such as motion rhythms that require long-term monitoring, the development of corrosion-resistant equipment materials will be another problem that needs to be overcome in future development.

## 6. Other Methods

In addition to the methods mentioned above, other technologies have been used to monitor behaviors and may be feasible alternatives, although there is no large-scale application.

The information collected by using a single technology is insufficient. In order to obtain more comprehensive and accurate behavioral information, researchers are trying to simultaneously use different technologies to obtain crustacean behavioral information from multiple angles. The combination of acoustic technology and sensor technology can yield behavior information from multiple angles. The technical fusion of acoustics and sensors can not only be used without obstacles in muddy underwater environments, but there is also an obvious absolute correspondence between the sound frequency of crustacean and the motion acceleration [127]. Therefore, it is feasible to use information fusion technology to make up for the blind spot of a single technology. This also provides a favorable theoretical basis for future large-scale research into information fusion technology in crustacean behavior monitoring.

Radio tag technology can also be used to quantify the behavioral characteristics of crustaceans; it transmits individual information to a receiving station or monitoring center. An RFID tag consists of a tiny radio transponder. When triggered by an electromagnetic interrogation pulse from a nearby RFID reader device, the tag transmits digital data, usually an identifying inventory number, back to the reader [103]. Radio tags are cheaper than acoustic tags and can be used to develop a low-cost real-time tracking system. However, tag loss during molting of the exoskeleton is the main difficulty and challenge of labeling technology to monitor crustacean behavior [137]. Therefore, the invention of internal elastomer tags could provide a new solution for the fixation of the label, and these tags would likely have large-scale applications in commercial fisheries in the future.

## 7. Challenges and Future Perspectives

The acquisition of crustacean behavior information is critical because it helps fishermen to know the behavior state in time, for applications such as grasping the best harvesting location according to the seasonal movement, and timely adjustment of the most suitable environmental parameters of crustaceans in the breeding period to provide a reference for obtaining the maximum welfare harvest. However, automatic monitoring of crustacean behavior is very difficult and challenging. One of the major reasons is that crustaceans are sensitive and translucent, and while monitoring behaviors the free movement of the crustacean should be ensured, which limits the application of many methods. Another reason is that the environmental characteristics of aquaculture are not conducive to crustacean behavior monitoring, such as low visibility, poor optical path through biofouling on optical systems, impossibility to discriminate individual animals, noise interference from apparatuses, inaccurate accelerometers, and electronic sensors being disturbed by electric fields. Manual monitoring is often ineffective, expensive, and damaging. With the development of advanced automation technologies such as machine vision, acoustics, and sensors there is significant potential to improve the precision of crustacean farming. However, the unique defect of each technology is also objective. The technical difficulties that need to be solved urgently include the substantive damage caused by sensors to crustaceans, how to move beyond single to multiple technology approaches, the low degree of automation, and the weak tracking ability of individual shrimp. Therefore, we propose future development trends in crustacean behavior monitoring to improve the level of precision aquaculture.

(1) It is necessary to expand and improve the application of imaging technology in aquaculture, which is suitable for crustacean breeding environments with low visibility and high density. In future studies, multiple types of imaging technologies can be used for behavior monitoring in aquaculture, moving beyond just to infrared imaging and RGB imaging. Microwave technology has been widely used in underwater imaging. Digital holography is one of the most advanced technologies used for monitoring aquatic animals. Therefore, to avoid the interference caused by the turbid water quality, microwave technology and digital holography can be used to monitor the behavior of crustaceans in the turbid water environment.

(2) Most machine-vision-based behavior monitoring uses planar images for analysis. Underwater 3D technology can conveniently obtain 3D coordinate information of crustaceans, which makes it easier to track individual crustaceans, improving the monitoring accuracy of the movement rhythm. Real-time aquatic behavior monitoring will support improved aquaculture management, welfare, and policy interventions.

(3) Deep learning (DL) is an algorithm that is highly suitable for underwater recognition. Performance comparisons with traditional methods based on manually extracted features indicate that the greatest contribution of DL is its ability to automatically extract features. Moreover, DL can also output high-precision processing results. A rapid, low-cost deep learning system would be highly suitable for the identification of individual crustaceans in a high-density stocking environment. Therefore, deep learning technology can be used to develop non-invasive, reproducible, and automated individual crustacean tracking and behavior monitoring.

(4) The combination of multiple technologies has been preliminarily explored in crustacean behavior monitoring. However, these electronic monitoring devices are inevitably affected by electric fields and accuracy. Therefore, a non-invasive method that combines multiple technologies has greater potential. For example, information fusion technology based on images and sensors is formed to solve the problem of a single device being affected by the environment and failure.

(5) Currently, the acoustic technology behavior monitoring method is seriously disturbed by noise. In addition to reducing the noise of equipment in the aquaculture environment as much as possible, the ability of acoustic technology should also be improved. Big data technology can efficiently analyze more data collected in one area or data collected across a larger area more frequently; fishermen will be able to determine changes in acoustic patterns more readily and compare them to other environmental data to provide a holistic understanding of crustaceans.

## 8. Conclusions

Over the past three decades, researchers have developed various automatic techniques and methods to monitor crustacean behaviors. This paper reviews current research concerning intelligent crustacean behavior monitoring, including acoustics, machine vision, sensors, and other emerging options. Based on an extensive analysis of the literature, Table 4 summarizes the advantages and disadvantages of various monitoring technologies and their wide range of applications, which could help provide the most suitable behavior monitoring means for different aquaculture environments. As a large-scale application technology, acoustics are not affected by water turbidity and they can work well in almost invisible conditions. Therefore, acoustic technology is more suitable than other methods for use in low visibility environments. However, their non-reusability, high cost, and noise interference limit their application in aquaculture. Compared with acoustics technology, machine vision is objective, repeatable, inexpensive, and not affected by noise; it can identify crustacean behavior remotely without causing damage or stress to the crustacean. The application of machine vision is limited by water surface reflectivity and low image quality. This problem can be solved by using near-infrared machine vision as its imaging quality is not affected by the intensity of visible light. In addition, it is necessary to develop more general sensors for a variety of crustaceans. The advantage of sensors is that they are inexpensive and highly accurate. However, they only work for large fish; the stress and damage caused by sensors on small aquatic animals limits their current development and applications. With the increasing diffusion of automation technology aquaculture in future, it can be expected that improved algorithms and new software will be developed for intelligent crustacean behavior monitoring to realize automatic aquaculture, and even unmanned fisheries.

## Figures and Tables

**Figure 1 animals-11-02709-f001:**
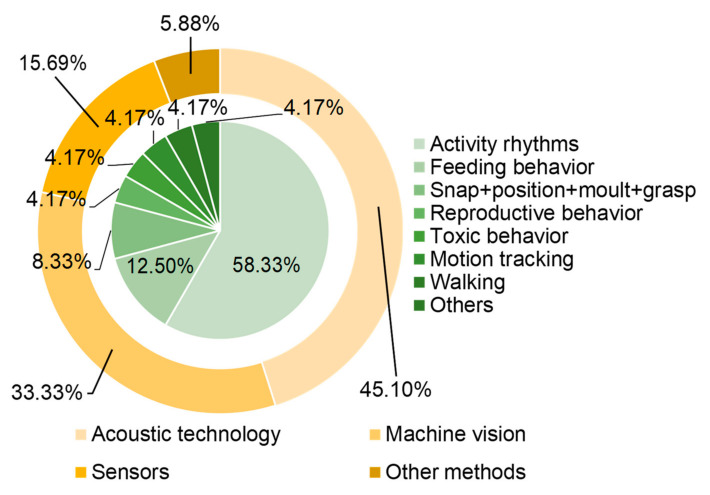
Number of papers with different methods and monitoring behaviors.

**Figure 2 animals-11-02709-f002:**
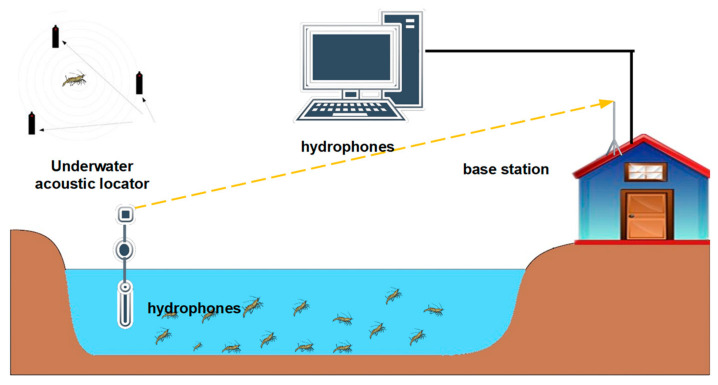
Acoustic technology overview.

**Figure 3 animals-11-02709-f003:**
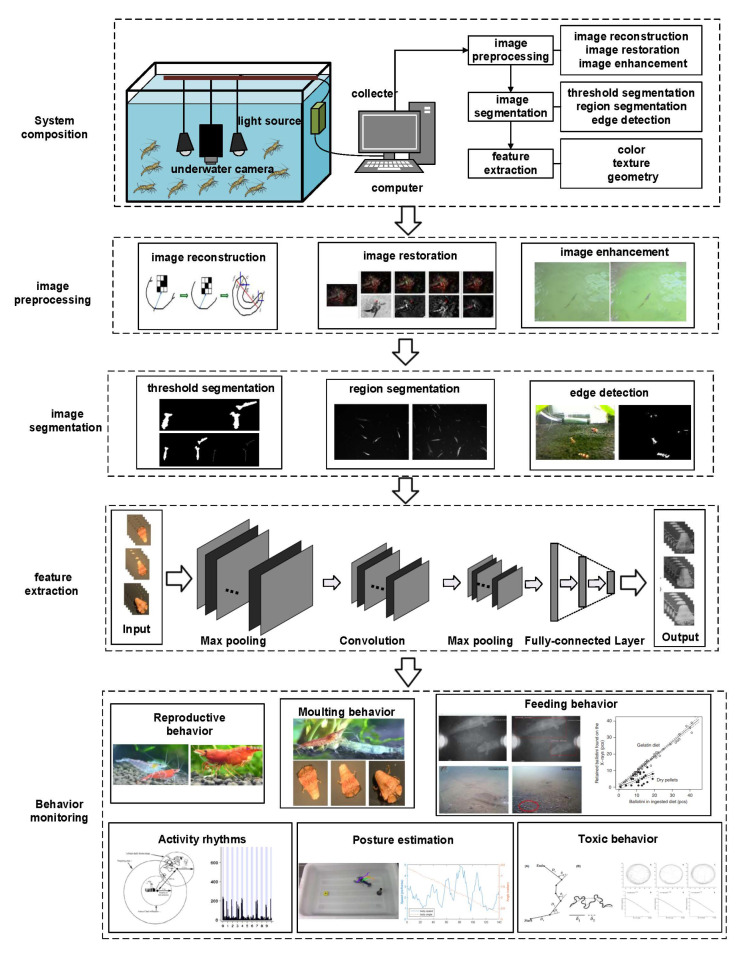
System structure and monitoring flow chart.

**Figure 4 animals-11-02709-f004:**
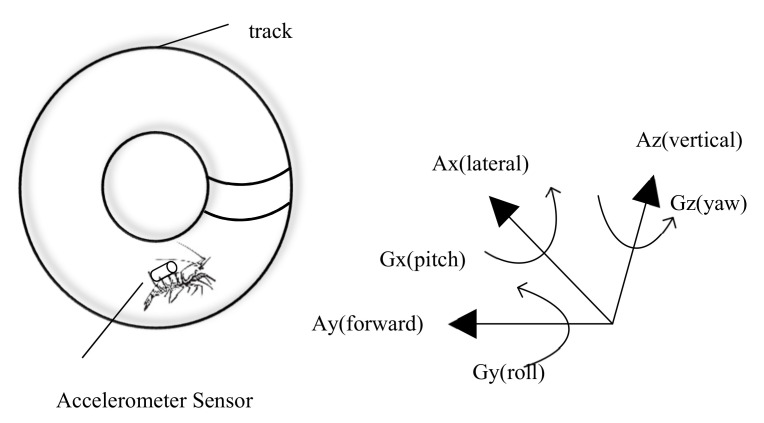
Crustacean behavior recognition based on an accelerometer.

**Figure 5 animals-11-02709-f005:**
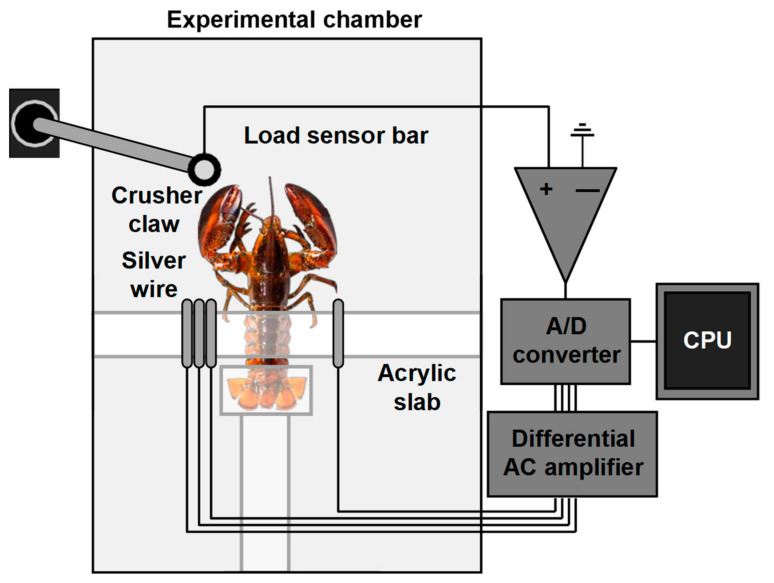
Crustacean behavior monitoring based on EMG.

**Table 1 animals-11-02709-t001:** Detailed information for behavior monitoring by acoustic technology.

Technology	Species	Application	Results or Accuracy	Culture Model	Acoustic Features/Principle	Reference
Passive acoustics	Tiger prawns	Feeding	R^2^ = 0.95 and R^2^ = 0.96	Tank and pond	3 kHz–7.6 kHz	[53]
	Prawn		Confidence intervals: 98.4 ± 0.6	Pond	51.2 kHz	[64]
	Family Alpheidae	Snap	R = 0.71–0.92	West Bay Marine Reserve	1.5–20 kHz	[21]
	Red swamp crayfish	Intraspecific interactions and activities	45% and *p* < 0.0001	Tank and natural environment	Peak frequency = 28 kHz Bandwidth RMS = 20 kHz	[65]
	European lobster	Seasonal activity	*p* < 0.05	Site	Vemco 12 VR^2^W	[66]
	Japanese spiny lobster	Movement		Island	0.04–21 kHz	[67]
Acoustic telemetry Active acoustics	European spiny lobster	Home range	*p* < 0.001	Protected area	Ultrasonic telemetry	[78]
	American lobsters		(523.2 ± 78.1 m/day^−1^; r^2^ = 0.62, *p* = 0.0001)	Enclosure	VEMCO V8SC-2L	[75]
	Lobster		SE = 0.09, *p* = 0.02	Coast	Vemco V13P–L	[6]
	European spiny lobster		Ranged from 1629.3 to 8641.3 m^2^	Coast	Vemco V9P-1L 69 kHz	[76]
	Spiny lobster		923 versus 871 m/day,	Channel	Vemco V16 69 kHz	[31]
	American lobsters		51% moved <5 km, 19% moved 5–10 km, and 30% moved >10 km	Inshore	Vemco V13-1L 69 kHz	[83]
	Lobster		The mean daily home range (n = 18) was 1002.0 ± 195.7 m^2^ (mean ± SEM)	Castle	VRAP model, VEMCO	[84]
	American lobsters		Home ranges (≈27.4−111.6 m^2^)	Castle	VRAP	
	Lobster Jasus lalandii	Nomadic behavior	*p* = 0.0002	Aquarium	Vemco V8-2LR	[77]
	Lobster	Reproductive migrations	Three migrations per year by an individual female	Western Sambo Ecological Reserve	Vemco V16 69 kHz	[79]
	Lobster	Feeding	90% confidence level (*p* = 0.09, K–S test)	Field enclosure	Vemco VR2W 69 kHz	[51]
	Spider crab	Migratory patterns	70% recapture rate	Coast	VEMCO V16	[85]
	Norway lobsters		Error < 1 m	European waters	Vemco	[87]
	Lobster	Movement patterns	r^2^ = 0.82, DF = 70, *p* < 0.0001	2.5 km^2^ lobster	Vemco VRAP	[29]
	Spider crab		R = 0.353; R = 0.805	Coast	VEMCO Ltd.	[80]
	Blue crabs		R > 0.64; R = 0.71–0.97; R = 0.25–0.32	Coast	Tucson Arizona	[86]
	Lobsters; crab			Tank	VEMCO Ltd.	[81]
	Edible crab		*p* < 0.001–0.042	Coast	Vemco VR 60	[82]

**Table 2 animals-11-02709-t002:** Summary of methods based on machine vision.

Principle	Application	Species	Culture Model	Result/Accuracy	Advantage/Disadvantage	Reference
Visible light	Molting behavior	Bay Lobsters	Tank	98.61%	Requires few training images	[106]
	Posture estimation	Lobster	Laboratory		Simple algorithm	[18]
	Mating behavior	Shrimp	Tank	Standard deviation: 5.8 ± 1.3	Affected by distance between shrimp and the camera	[107]
	Toxic behavior	Shrimp	Tank	0.05 < *p* < 0.08	Low cost	[109]
		Crayfish	Tank	*p* < 0.001	Poor timeliness	[110]
	Daily locomotor activity	Norway Lobster	Tank	60% and 89.5%	For long-term monitoring	[123]
	Activity rhythms	Norway Lobster	Tank	*p* < 0.01	Faster detection	[124]
		Norway Lobster	Tank	range: 139.8–1917.1 cm	Low cost	[114]
		Norway lobster	Tank	98.7% and 76.9% (*p* = 0.329)	Poor detection result of oblique camera	[12]
		Shrimp	Pond	Successfully detected	Function can be expanded	[113]
	Feeding	Shrimp	Shrimp farming	0.03–0.63	Need better image enhancement techniques	[114]
	Cleaning, food burying, feeding, moving, substrate, resting	Lobster	Artificial lobster cavity		Wide application range, can monitor multiple behaviors	[34]
	Burrowing, walking, resting, ventilating	Mud shrimp	Aquarium	82.7%	Expensive	[35]
	Motion tracking	Gammarus pulex	Tank		Flexible data analysis	[111]
		Crayfish	Tank	*p* = 0.05	Noninvasive	[112]
Near infrared	Activity rhythm	Norway lobster	Tank	*p* < 0.5	Noninvasive	[39]
X-ray	Feeding	Crayfish	Tank	92.8 ± 8.6%	Damaged	[121]

**Table 3 animals-11-02709-t003:** Summary of methods based on movement sensors.

Technology	Application	Species	Culture Model	Result/Accuracy	Advantages/Disadvantages	Reference
Acceleration sensor	Diel activity patterns	Slipper lobsters	Tanks	R^2^ = 0.898; *p* ≤ 0.01	Direct detection, high accuracy/contact, damage to shrimp body	[125]
		Spiny lobster	Tank	χ󠇕^2^ = 23.10, df = 2, *p* < 0.001		[127]
	Movement	Lobster	Laboratory	R^2^ = 0.63, *p* < 0.001		[128]
		Lobster	Laboratory	accuracy > 90%		[14]
Electromyogram (EMG) transmitter	Walking	Crayfish	Tank	*p* < 0.05	Directly reflects hunger levels; damage to fish body	[130]
	Leg movement	American lobster	Laboratory	*p* < 0.05		[133]
	Circadian rhythms	Crayfish	Laboratory	*p* < 0.05		[135]
	Grasping behavior	American lobster	Laboratory	*p* = 3.099 × 10^−9^, <0.05		[134]

**Table 4 animals-11-02709-t004:** Advantages and disadvantages of different automatic monitoring methods.

Technology	Advantages	Disadvantage	Application and Acceptance Level
Acoustic			
Passive acoustics	Rapid and noninvasive, useful in water with low ambient light	Many other forms of impulse, large amount of data, difficult to distinguish	Large-volume aquaculture model, such as a tank system,
Acoustic telemetry	High precision and high data resolution, regardless of turbidity and light	Expensive, acoustic label is a contact technology, limited number of tracked shrimps	Small-scale farming
Machine vision			
Visible light	Low cost, repeatability, real time, noninvasive	Higher visibility requirements, susceptible to water turbidity	Good lighting condition
Invisible light	Regardless of visible light intensity, no calibration	The short penetration, refraction, and scattering of infrared rays	Laboratory conditions
Sensors			
Acceleration sensor	Regardless of turbidity and light, readily adaptable to field	Contact with shrimp, damage to shrimp body, susceptible to other parameters	Specific tank and pond
EMG	High precision, can determine whether the behavior is reflexive or spontaneous	Implanted in shrimp, cannot be used on small shrimp	Laboratory conditions

## Data Availability

No new data were created or analyzed in this study.

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
