# Peer review of "Automatic Monitoring of Relevant Behaviors for Crustacean Production in Aquaculture: A Review"

_animals, 2021, doi:10.3390/ani11092709_

Round 1

Reviewer 1 Report

I added some comments and questions to the manuscript that would be of interest for aquaculturists. An assessment of the most promising method would be helpful to direct research towards this technology. 

Author Response

General comments:

I added some comments and questions to the manuscript that would be of interest for aquaculturists. An assessment of the most promising method would be helpful to direct research towards this technology.

Response:

Thank you very much for your positive comments and approvals. We have repeatedly read and revised this paper following your valuable advice. The specific revisions can be seen in the revised paper, where all the revisions are in revised mode and highlighted in yellow.

  1. Animal health is a result of the appropriateness of the production environment. Behavioural changes are often linked to adverse environmental conditions. Behavioural observations may point on those reactions after they had emerged which is mostly a result of inappropriate aquaculture biotechnology and poorly automated systems. Thus, behavioural observation may a posteriori indicate changes in the production environment. The automation of aquaculture system (process technology) seems to be the solution to maintain appropriate living conditions?

Answer1:

- Thank you for your comment, we agree with your view. Behavior observation is a posteriori indicate changes in the production environment. Abnormal behavior is the key evidence of unsuitable living conditions. When abnormal behavior is observed, it also means that the water quality environment has changed, and the timeliness is poor. Normal behaviors such as feeding behavior and reproduction behavior occur, the living environment will not change. These behaviors are the key to assessing welfare. Therefore, monitoring abnormal behaviors (stress behavior) is only a reference for changes in the aquaculture environment, and a more effective method is aquaculture system (process technology). Normal behavior monitoring can provide a basis for practitioners to grasp the growth status of crustaceans. At your prompt, we revised lines 63-67 of the manuscript:

“Compared with the environmental parameter detection system, automatic behavior monitoring is a posteriori indicate, but it is very meaningful for welfare evaluation. In terms of feeding, moving, home range, and activity rhythms may grasp biological behavior information, real-time monitor animal health, and provide early warning of diseases.”

  1. how to assess individual behavior in a crowded, possible noisy aquaculture environment?

Answer2:

- Thank you for your comment, we reviewed the existing references, in a crowded, noisy aquaculture environment, it is still a difficult point to assess individual behavior. The application environment that can assess individual behavior is an environment where individual behavior signals can be obtained with high visibility, low density, and low environmental noise interference. Individual behavior recognition in extreme crowded and noisy environments has not yet been seen.

- At your prompt, we have added this information in the Challenges and future perspectives section, and suggest deep learning, bio-floc-technology, reducing the noise of equipment to solve this problem.

  1. Feed uptake follows internal and external stimuli. It seems appropriate to feed using a feeding table and schedule. Observation empower during the operation of aquaculture systems to stop the feeding of animals if appetite slow down?

Answer3:

- Thank you for your comment, we agree with your view. Feeding table and schedule are appropriate to feed. It can set the most suitable feeding amount and time according to the current environmental parameters, is an effective method to achieve precise feeding in aquaculture. When appetite slows down, observation empower during the operation of aquaculture systems will stop the feeding animals. Machine vision detection and sensor detection of feeding behavior usually set a threshold, and when it detects that the remaining amount of feed exceeds the threshold, the feeding is stopped. Though, this is inferior to the accuracy of using a feeding table and schedule, it can prevent accidents. For example, if most crustaceans stop feeding due to other reasons, automated behavior monitoring can stop feeding in time, while the feeding table and schedule will continue to feed according to the program. In order to avoid misunderstandings, we have revised lines 104-106 of the original manuscript:

“Feeding table and schedule is a common and accurate feeding method, but automated feeding behavior recognition also has great potential in determining when to start and stop feeding in order to improve feed conversion rate and reduce costs.”

  1. My own observation: shrimp carry feed items to their resting place to slowly ingest the feed. thus, a model based feeding probably works better?

Answer4:

- Thank you for your comment, we agree with your view. The most difficult aspect of shrimp feeding model establishment is that shrimp carry feed items to their resting place to slowly ingest the feed. This is also the main reason for the loss of a lot of feeding information. Insufficient feeding impede shrimp growth while excessive amounts of feed inputs lead to environmental degradation and unnecessary high investment. Establishing a complete feeding model that includes habitat feeding information will be helpful to solve this problem and has a better effect.

- At your prompt, in order to obtain the feeding information of shrimps in resting place, researchers can use multiple monitoring devices to obtain more detailed information. Therefore, the feeding model that includes the feeding information in the shelter will be more accurate and effective, even if the shrimp carry feed items to their resting place.

  1. A comprehensive inventory of stereotype behavior would certainly help to assess performance under aquaculture conditions. this is likely a task that requires machine learning and specialized sensor technology. but, at the end it is the task of biologists to gauge the importance of stereotype behavioral pattern before it can be used.

Answer5:

- Thank you for your comment, we agree with your view. A comprehensive inventory of stereotype behavior can help to assess performance under aquaculture conditions, biologists need to gauge the importance of stereotype behavioral pattern before it can be used. Stereotyped movement is used to describe physical movements that are both aimless and repetitive. Therefore, the movement rhythm information of crustaceans requires further judgment by biologists. If it is a regular seasonal movement, it will be of great help for choosing the correct location for fishing, but if it is an irregular movement, it will cause interference. Stereotyped behaviors cannot be used as a basis for judging the conditions of aquaculture, professional environmental parameter monitoring equipment is the solution to maintain the aquaculture environment. At your prompt, we modified lines 113-114 of the original manuscript, and the revised content is as follows:

“As a basis for welfare assessment, the rhythms of movement behavior can only be used to help choose the correct location for fishing, are not the core of assessing performance under aquaculture conditions.”

  1. does this work in an crowded aquaculture environment?

Answer6:

- Thank you for your comment. Passive acoustic behavior detection technology is effective in crowded aquaculture environment. References [55-63] are the theoretical basis for the sound production of crustaceans in some behaviors. Reference [64] records the signatures of the feeding sound of a set of 2400 prawn in a farm pond. The author filtered out noise and the feeding event of prawns was successfully identified. Reference [65] used a remote acoustic recording station to study the circadian underwater acoustic activity of the crayfish and to assess the acoustic features of the signals in a lake. Reference [66] studied movement and activity patterns of European lobsters in natural environments. Reference [67] explored the frequency of stridulating sounds from Japanese spiny lobster’s in crowded cage aquaculture. Reference [21] explored the relationship between group snapping shrimp sound and seasonal movement. There may be different behaviors in the crowded aquaculture environment, and the sound frequencies of different behaviors are different. Researchers can identify the behaviors of interest based on the spectral characteristics of the sound. The recognized behavior sounds are also group, and it is not possible to assess individual behavior. This result can be used to estimate the strength of the behavior and the number of individual crustaceans. The above references are all about the application of passive acoustic technology in the environment of crowded aquaculture. These scholars have realized the monitoring of crustacean group behaviors by understanding the sound frequency of different behaviors. Therefore, passive acoustic behavior monitoring can be used in group aquaculture environments.

  1. (65) … „The tests were performed in the laboratory under controlled conditions and free of noise or any other acoustic interference to capture the sound produced by the individuals with no spectral overlap. “ this setup is far out of aquaculture

Answer7:

- Thanks for your comment, we agree with your view, References [65] was performed in the transparent glass aquarium (48×23.8×38 cm), the test conditions can be controlled, no noise or any other acoustic interference to capture the sound produced by the individuals with no spectral overlap. This is an ideal experimental environment, which is far out of the actual aquaculture environment. At your prompt, we deleted the reference [65] related content in the manuscript, which is inconsistent with the content of the manuscript.

  1. what is great promise? is it possible to filter out noise from the sound produced by crustacean?

Answer8:

- Thanks for your comment. We agree with your suggestion. The great promise means that the new machine learning method can recognize the behavioral sounds of interest in a very noisy environment, which means that these noises can be filtered out. It works as follows: machine learning methods such as neural networks extract time-frequency map features of behavioral sound signals, and establish an automatic sound recognition model to recognize behavioral sounds.

To better explain the role of machine learning in sound processing, we modified the original sentence to: “The signal detection method based on machine learning can extract the time-frequency characteristics of the sound and filter out the interference of noisy sounds.”

  1. (81) Lobster are of different size and are living as single animals compared to aquaculture with 10000 of animals in one tank. This situation clearly does not support any natural behavior which would be the appropriate approach to improve animal welfare.

Answer9:

- Thanks for your comment, we agree with your view, the experiment in reference [81] was carried out in a large (50 m × 50 m) underwater enclosure (mesocosm), experimental lobsters are of different size and are living as single animals. We did not compare the experimental environment with the actual aquaculture environment. Therefore, the behavior in this situation is meaningless to improve animal welfare. - At your prompt, we deleted the relevant content of reference [81].

  1. what about 3D technology to assess the position in the three dimensional living space? is it necessary or not?

Answer10:

- Thanks for your comment, we believe that the application of 3D technology is still necessary in aquaculture. We explained why 3D technology is necessary and how 3D technology obtains spatial location information. A 3-dimensional (3D) approach improves the capacity to accurately measure the position of moving animals in space, and plays a significant role in accurately acquiring the movement of crustaceans. 2D images can only obtain the movement information of crustaceans on a plane. In fact, animals live in three-dimensional space. Therefore, 2D technology has great flaws in the calculation of activity volume and activity.

- 3D technology closely copies how our eyes work to give us accurate, real-time depth perception. It achieves this by using two sensors a set distance apart to triangulate similar pixels from both 2D planes. Each pixel in a digital camera image collects light that reaches the camera along a 3D ray. If a feature in the world can be identified as a pixel location in an image, we know that this feature lies on the 3D ray associated with that pixel. If we use multiple cameras, we can obtain multiple rays. Finding where these rays intersect tells us the 3D location of an object and its features. Through basic triangulation of pixels and ray intersections, we can determine the 3D location of crustaceans.

- At your prompt, we also revised the application of 3D technology in the acquisition of location information in aquaculture in the future development trend. The revised content is as follows:

“Most machine vision-based behavior monitoring uses planar images for analysis. Underwater 3D technology can conveniently obtain 3D coordinate information of crustacean, which makes it easier to track individual crustacean, improves the monitoring accuracy of the movement rhythm. Real-time aquatic behavior monitoring will support improved aquaculture management, welfare, and policy interventions.”

  1. in marine shrimp aquaculture two ways of husbandry needs to be discussed:

(1) intensive high density production in crowded tanks is causing extremely high turbidity (bio-floc-technology). I it possible to discriminate between individual animals?

(2) are in biological (behavioral) more appropriate production systems where animals are able to hide in structures/sediments as part of their normal behavior multiple video systems necessary to follow the behavior of individual animals?

Answer11:

- Thanks for your comment.

(1) We agree with your view, machine vision is still unable to discriminate individual crustaceans in intensive high density aquaculture environment. The main reasons are turbid water and low visibility. References [12], [34,35], [93], [106,107], [109-113] all realized the individual identification of crustaceans in a clearer underwater environment. In order to discriminate individual crustacean in a high-density aquaculture environment. One method is to improve water quality and increase visibility. At your prompt, bio-floc-technology is an effective method to improve water quality, waste treatment and disease prevention in intensive aquaculture systems. Another method is to improve image processing functions to realize the application of deep learning technology to identify individuals. This method was also mentioned in the challenges and future perspectives section. Therefore, the combination of machine vision technology and bio-floc-technology will bring more possibilities for individual identification in high-density crustacean crowded environments. We added the development direction of the combination of bio-floc-technology and machine vision technology in the last paragraph of Chapter 4: “The combined use of bio-floc-technology and machine vision technology can make it possible to identify individual animal in intensive high density environment.”

(2) It is necessary to follow the behavior of individual animals multiple video systems. Crustaceans are able to hide in shelters, it is also one part of their normal behavior. Therefore, it is necessary to place an additional video system in the shelter, which will improve the behavior information of crustaceans. At your prompt, we added the development direction of multiple video systems in the last paragraph of Chapter 4: “The multiple video systems can collect more behavior information of crustaceans.”

  1. shelter, seem to be necessary in marine shrimp aquaculture to better survival and sustainability of production (loss is a waste of resource).

Answer12:

- Thanks for your comment, we agree with your view, shelter is necessary in marine shrimp aquaculture to better survival and sustainability of production. There is a lot of important information about the crustaceans hiding in the shelter, which is exactly the problem that needs to be solved in the future. At your prompt, we proposed in answer 11 that multiple monitoring devices can be used to obtain information on the behavior of the shelter.

  1. works with large animals?

Answer13:

- Thanks for your comment, we agree with your view. The acceleration sensor usually needs to be fixed on the crustacean, so it will apply pressure to animals. Although the size and weight of acceleration sensors are getting smaller than before, most species of crustaceans are small. The pressure effect of the sensor on crustaceans will be also great, especially for small species. Therefore, accelerometry appears to be a suitable method for assessment of movement patterns and distance traveled by animals above a certain size, cannot be used on small crustacean (e.g., krill and fairy shrimp). The literatures reviewed in the manuscript were also about the behavior monitoring of larger crustaceans by accelerometers. We have listed the parameters of the accelerometer sensors used in some references, which are as follows:

- Reference [13], tri-axial accelerometer (CEFAS, Lowestoft, England. Model G6A, 2.3 g in water/7.3 g in air, 40 × 28 × 15 mm), experimental animal: lobster, males 718.1 ± 336.6 g and females 770.5 ± 169.2 g.

- Reference [123], accelerometer (HOBO Pendant G, model: UA-004-64, Onset Computer Corp, Bourne, MA, USA) weight 18 g (0.6 oz), dimensions 58 x 33 x 23 mm (2.3 x 1.3 x 0.9 inches) experimental animal: slipper lobsters, average length: 94.6±2.0 mm.

- Reference [126], V13AP tag and HOBO weight <12 g, experimental animal: lobsters, range: 81–87 mm.

- From the above references, accelerometers are all applied to larger crustaceans such as lobsters. The accelerometer cannot be used on crustaceans that are smaller than the accelerometer. Therefore, the size and weight of the acceleration sensor are the main reasons that limit its application to small crustacean. At present, it can only be used in large crustaceans. In the future, the improvement and application of implanted sensors can promote the tracking and monitoring of small crustaceans.

  1. Jury et al 2018 (doi:10.5343/ bms.2017.1117): the correlation between video observation of movement and relative accelerometer activity seems to be rather weak.

Answer14:

- Thanks for your comment, we agree with your view, the correlation between video observation of movement and relative accelerometer activity is rather weak. References [126] (Jury et al 2018 doi:10.5343/ bms.2017.1117) shows that the correlation between video distance and the output of the two acceleration sensors is: r2 = 0.53, P < 0.05; r2 = 0.63, P < 0.001. References [119] there was a relatively good relationship between the activity measured using accelerometry and video analysis (r2 = 0.898, P < 0.0001). R2 value means how much variation is explained by model. So 0.1 r2 means that the model explains 10% of variation within the data. The greater r2 the better the model. Whereas p-value means the F statistic hypothesis testing of the "fit of the intercept-only model and model are equal". Therefore, if the p-value is less than the significance level (usually 0.05) then models fit the data well.

It is possible to get a significant p-value with a low R-squared value. This often happens when there is a lot of variability in the dependent variable, but there are enough data. The low r2 and low p-value in reference [126] (p <0.05) means that there is no good relationship between video distance and acceleration sensors, but it is important (better than no model). The high r2 and low p-value in reference [123] means the model explains a lot of variation within the data and is significant (best scenario). We did have flaws in the presentation of the manuscript.

- At your prompt, we re-summarized and evaluated references [123] and references [126], and revised lines 446-452 of the original manuscript. The revised content is:

“The collected accelerometer outputs can be converted into the distances moved per unit time and scholars can estimate the distance moved by shrimp in a period of time according to this method to an extent that is statistically significant, that is P <0.005 [123,126]. However, the correlation between the movement and accelerometer is uncertain. Jury, S.H et al., obtained the value of r2 between the video activity and accelerometer activity of 0.898. The activity was defined as forward, backward, or sideways locomotion (>2 cm) for each lobster [126]. Goldstein, J.S. et al., obtained the value of r2 between the video distance and acceleration of 0.53 and 0.63 [123]. These results indicate that accelerometer only can estimate activity, and it is still not accurate enough for more demanding distance calculations.”

This is just as you mentioned that the direct observation of by accelerometers are probably not accurate enough is a challenge at this stage. Thanks for your comment.

  1. does this contribute to the protection of wild stock or does it support the overexploitation?

Answer15:

- Thanks for your comment. This sentence intends to explain that the information of behavior can provide the breeders with the status of crustaceans, such as whether they have entered the breeding period, whether they have stopped feeding, whether there has been seasonal movement, etc. I very much agree with your point of view. We can also use this information to protect wild crustaceans. For example, when large-scale seasonal movements occur, fishmen can timely adjust the environment of the migration locations.

At your prompt, we have revised the original sentence to: “The acquisition of crustacean behavior information is critical because it helps fishermen to know the behavior state in time, such as grasping the best harvesting location according to the seasonal movement; Timely adjust the most suitable environmental parameters of crustaceans in the breeding period to provide a reference for obtaining the maximum welfare harvest.”

  1. Monitoring of animal crustacean behavior in aquaculture seems to be hindered by

(1)low visibility due to poor water quality,

(2)poor optical path through biofouling on optical systems,

(3)the impossibility to discriminate between individual animals because of the stocking density,

(4)noise generated by apparatuses such as aerators, pumps, worker’s activity,

(5)direct observation by accelerometers are probably not accurate enough and not applicable in aquaculture.

(6)electric sensors may be disturbed by electric fields generated through the employed apparatuses.

Answer16:

- Thanks for your comment, we agree with your view, we have detailedly revised our manuscript. In the current difficulties and challenges, we added current problems such as low visibility, poor optical path, impossibility to discriminate individual animals, noise interference, inaccurate accelerometers, and electric sensors may be disturbed. And in the future development trends and suggestions, the corresponding solutions have been modified. Lines 537-542 of the original content was revised as follows:

“One of the major reasons is that crustacean are sensitive and translucent, and monitoring behaviors the free movement of crustacean should be ensured, that limits the application of many methods. Another reason is that the environmental characteristics of aquaculture are not conducive to crustacean behavior monitoring, such as low visibility, poor optical path through biofouling on optical systems, impossibility to discriminate individual animals, noise interference from apparatuses, inaccurate accelerometers and electronic sensor is disturbed by electric field.”

At your prompt, we rewrote the future development trends, Lines 551-580 of the original content was revised as follows:

(1) It is necessary to expand and improve the application of imaging technology in aquaculture, which is suitable for crustacean breeding environments with low visibility and high density. In future studies, multiple types of imaging technologies can be used for behavior monitoring in aquaculture, moving beyond just to infrared imaging and RGB imaging. Microwave technology has been widely used in underwater imaging. Digital holography is one of the most advanced technology used for monitoring aquatic animals. Therefore, to avoid the interference caused by the turbid water quality, Microwave technology and digital holography can be used to monitor the behavior of crustaceans in the turbid water environment.

(2) Deep learning (DL) is an algorithm that is highly extremely suitable for underwater recognition. Performance comparisons with traditional methods based on manually extracted features indicate that the greatest contribution of DL is its ability to automatically extract features. Moreover, DL can also output high-precision processing results. A rapid, low-cost deep learning system would be highly suitable for the identification of individual crustaceans in high-density stocking environment. Therefore, deep learning technology can be used to develop non-invasive, reproducible, and automated individual crustacean tracking and behavior monitoring.

(3) The combination of multiple technologies has been preliminarily explored in crustacean behavior monitoring. However, these electronic monitoring devices are inevitably affected by electric fields and accuracy. Therefore, a non-invasive method that combine multiple technologies have greater potential. For example, information fusion technology based on images and sensors is formed to solve the problem of a single device affected by the environment and failure.

(4) Currently, acoustic technology behavior monitoring method is seriously disturbed by noise. In addition to reducing the noise of equipment in the aquaculture environment as much as possible, the ability of acoustic technology should also be improved. Big data technology can efficiently analyse more data collected in one area or data collected across a larger area more frequently, fishmen will be able to determine changes in acoustic patterns more readily and compare them to other environmental data to provide a holistic understanding of crustacean.

  1. Deep learning needs to support the observation of individual animals if behavior is the direct measure of welfare, activity, feeding, ….

Answer17:

- Thanks for your comment, we agree with your view, behavior is directly related to the measurement of welfare, activity, feeding, etc. We have revised the section about the future development trend of deep learning, and will focus on using deep learning technology to identify individual crustaceans in high-density farming. The revised content is as follows:

(2) Deep learning (DL) is an algorithm that is highly extremely suitable for underwater recognition. Performance comparisons with traditional methods based on manually extracted features indicate that the greatest contribution of DL is its ability to automatically extract features. Moreover, DL can also output high-precision processing results. A rapid, low-cost deep learning system would be highly suitable for the identification of individual crustaceans in high-density stocking environment. Therefore, deep learning technology can be used to develop non-invasive, reproducible, and automated individual crustacean tracking and behavior monitoring.

  1. noise in aquaculture environments generated by pumps, aerators needs to be eliminated … noise cancelling technology?

Answer18:

- Thanks for your comment, we agree with your view, the noise produced by pumps, aerators are the main difficulty in the application of acoustic technology in aquaculture.

- At your prompt, we revised content to: “However, their non-reusability, high cost and noise interference limit their application in aquaculture. Compared with acoustics technology, machine vision is objective, repeatable, inexpensive, and not affected by noise; it can identify crustacean behavior remotely without causing damage or stress to the crustacean.”

Reviewer 2 Report

I think this review is generally well-organized and explaining recent technologies about automatic monitoring system for crustaceans. My concerns are as follows.

Digital holography is one of the most advanced technology used for monitoring microcrustaceans (eg. Sunayama et al. 2019, https://doi.org/10.1007/s10043-019-00558-8; MacNeil et al. 2021, https://doi.org/10.1186/s12862-021-01839-0). Even for large species, shrimps and crabs, it can be used for larvae. Authors should mention this technology.

Characters and pictures in figures are too small and difficult to understand what they mean, especially in Figure 1 and 3. These figures should be reconstructed radically.

Author Response

General comments:

Comments and Suggestions for Authors

I think this review is generally well-organized and explaining recent technologies about automatic monitoring system for crustaceans. My concerns are as follows.

Response:

Thank you very much for your positive comments and approvals. We have repeatedly read and revised this paper following your valuable advice. The specific revisions can be seen in the revised paper, where all the revisions are in revised mode and highlighted in yellow.

  1. Digital holography is one of the most advanced technology used for monitoring microcrustaceans (eg. Sunayama et al. 2019, https://doi.org/10.1007/s10043-019-00558-8; MacNeil et al. 2021, https://doi.org/10.1186/s12862-021-01839-0). Even for large species, shrimps and crabs, it can be used for larvae. Authors should mention this technology.

Answer1:

- Thanks for your comment, we agree with your view. Digital holography refers to the acquisition and processing of holograms with a CCD camera or a similar device. Digital holography offers a means of measuring optical phase data and typically delivers three-dimensional surface or optical thickness images. It is one of the most advanced technology used for monitoring microcrustaceans. We searched the application references of digital holography in aquaculture, including the two references you mentioned. The main applications of digital holography are aquatic particle characterization and plankton distributions and patchiness. References [Sunayama et al. 2019] measured the size of Daphnia pulex; References [MacNeil et al. 2021] is about plankton classification. The current application objects of digital holography are mainly particles and plankton. At your prompt, although digital holographic has not yet been widely applied to crustaceans, it has great potential for monitoring animals, especially individual behavior monitoring. Therefore, in the challenges and future perspectives, we proposed this technology as an effective method to solve the identification of individual crustaceans in a high-density aquaculture environment. The revised content is as follows:

“It is necessary to expand and improve the application of imaging technology in aquaculture, which is suitable for crustacean breeding environments with low visibility and high density. In future studies, multiple types of imaging technologies can be used for behavior monitoring in aquaculture, moving beyond just to infrared imaging and RGB imaging. Microwave technology has been widely used in underwater imaging. Digital holography is one of the most advanced technology used for monitoring aquatic animals. Therefore, to avoid the interference caused by the turbid water quality, Microwave technology and digital holography can be used to monitor the behavior of crustaceans in the turbid water environment.”

  1. Characters and pictures in figures are too small and difficult to understand what they mean, especially in Figure 1 and 3. These figures should be reconstructed radically.

Answer2:

Thanks for your comment, we agree with your view. At your prompt, we have reconstructed Figure 1 and 3 radically and improved the clarity of all the pictures in the manuscript. The revised pictures are as follows. We have attached vector diagram pictures in the revised manuscript. If the clarity of these pictures is still poor, we can upload the original pictures. Please see the attachment.
